# 3D-Printing of Capsule Devices as Compartmentalization Tools for Supported Reagents in the Search of Antiproliferative Isatins

**DOI:** 10.3390/ph16020310

**Published:** 2023-02-16

**Authors:** Camilla Malatini, Carlos Carbajales, Mariángel Luna, Osvaldo Beltrán, Manuel Amorín, Christian F. Masaguer, José M. Blanco, Silvia Barbosa, Pablo Taboada, Alberto Coelho

**Affiliations:** 1Departamento de Química Orgánica, Facultad de Farmacia, Universidade de Santiago de Compostela, 15782 Santiago de Compostela, Spain; 2Departamento de Física de la Materia Condensada, Facultad de Física, Universidad de Santiago de Compostela, CP 15782 Santiago de Compostela, Spain

**Keywords:** 3D printing, catalyst compartmentation, polypropylene capsule, supported reagents, isatins, anticancer

## Abstract

The application of high throughput synthesis methodologies in the generation of active pharmaceutical ingredients (APIs) currently requires the use of automated and easily scalable systems, easy dispensing of supported reagents in solution phase organic synthesis (SPOS), and elimination of purification and extraction steps. The recyclability and recoverability of supported reagents and/or catalysts in a rapid and individualized manner is a challenge in the pharmaceutical industry. This objective can be achieved through a suitable compartmentalization of these pulverulent reagents in suitable devices for it. This work deals with the use of customized polypropylene permeable-capsule devices manufactured by 3D printing, using the fused deposition modeling (FDM) technique, adaptable to any type of flask or reactor. The capsules fabricated in this work were easily loaded “in one step” with polymeric reagents for use as scavengers of isocyanides in the work-up process of Ugi multicomponent reactions or as compartmentalized and reusable catalysts in copper-catalyzed cycloadditions (CuAAC) or Heck palladium catalyzed cross-coupling reactions (PCCCRs). The reaction products are different series of diversely substituted isatins, which were tested in cancerous cervical HeLa and murine 3T3 Balb fibroblast cells, obtaining potent antiproliferative activity. This work demonstrates the applicability of 3D printing in chemical processes to obtain anticancer APIs.

## 1. Introduction

Three-dimensional printing is a revolutionary technology in many scientific fields today. Some of its scopes have a great impact on health sciences and particularly on medicine [1,2]. The application of 3D printing in the field of chemistry in general and specifically in pharmaceutical chemistry has been driven mainly by the initial work of Cronin and collaborators [3,4] through the design of polypropylene reactors with different shapes and compartments that make the concept of automated synthesis of “drugs on demand” a reality [5]. These initial works highlighted concepts of extraordinary importance today, such as easy work up or to facilitate the dispensing of reagents by automated robotic systems connected to multimedia systems. Collectively, behind these studies is the idea of “democratizing” and digitizing the syntheses and their scale up in many laboratories [6]. This goal is even more feasible if we consider the considerable technological advancement of 3D printers in recent years [7,8]. On the other hand, there is currently an increasingly deep concern for the development of recyclable and/or reusable materials and devices and, particularly in the field of green chemistry, regarding the development of heterogeneous catalysts or effective polymeric reagents [9]. Good examples of this class of reagents are polymeric ones containing gold, palladium, platinum, ruthenium, or copper metal and even scavenger-type polymers for toxic substances, among others [10,11,12,13,14]. Particularly powerful are the copper-catalyzed alkyne-azide cycloadditions (CuAAC, the flagship of the click chemistry) or the palladium-catalyzed cross-coupling reactions (PCCCRs) [15,16,17]. However, many of the heterogeneous catalysts have a very small particle size (in general, metal nanoparticles [18], silicas [19], and mainly polystyrene-based polymers [20]). In the case of polymeric reagents, the size range is highly variable, but many are found in diameters of a few hundred microns thick. This implies that the solid reagent or heterogeneous catalyst, which in many cases is expensive, must be separated from the reaction medium by filtration to be recycled or reused. In many cases, it would be interesting to be able to use two solid catalysts simultaneously (for use in tandem processes [21], cooperative bimetallic catalysis [22,23], multicatalytic one-pot procedures [24], etc.) enabling their reuse separately. Therefore, from a practical point of view, the compartmentalization of catalysts is an interesting objective in the coming years.

The concept of catalyst compartmentalization [25,26] was promoted by Houghten in the 80s, mainly through the “tea bag technique” [27] and was applied to combinatorial chemistry. Another impulse in this sense was contributed by Jones and other groups, using magnetic nanoparticles, easily extractable from the reaction medium and allowing consecutive “one pot” functionalizations with incompatible reagents, such as acids and bases [28,29,30]. In any case, the use of solid materials in solution phase organic synthesis (SPOS) depends on their adaptability and chemical resistance to the reaction mixture, so the selection of the material is very important. Our research group has carried out multicomponent heterogeneous multicatalytic syntheses (MCRs) including 3D-printed ceramic monoliths in the reaction cocktails [31], in which two compartmentalized catalysts in each monolith can carry out these kinds of transformations. A crucial aspect in these procedures is that the metal species remains immobilized on the support or heterogeneous catalytic material, allowing its reuse and, at the same time, avoiding the contamination of the reaction medium with the metallic species and, therefore, the drug. This aspect is fundamental if we consider the strict controls imposed by international agencies for evaluation of medicinal products regarding the presence of these metals in the final drugs [32].

The isatin (1H-indole-2,3-dione) nucleus is a privileged scaffold in medicinal chemistry. The versatility of isatin’s molecular architecture makes it an ideal scaffold for structural modification and functionalization as evidenced by the fact that many isatin derivatives exhibit a broad range of biological activities [33,34,35,36,37,38,39]. Particularly comprehensive are the reviews by the groups of Shankaraiah and Costa Ferreira about the anti-neoplastic activity of variously substituted isatins [40,41]. Among the structural modifications carried out on the scaffold for the search of new antitumoral chemodrugs, those carried out on position 3 of the oxindole stand out, either through the formation of 3-imino, 3-hydrazine-ylidene, and especially 3-alkylideneindolin-2-one type bonds or through the development of Ugi-type multicomponent reactions (MCRs) [42]. This has given rise to very powerful and effective drugs such as the experimental drug Orantinib [43,44], the commercialized Sunitinib [45] or Nintedanib [46]. The transformations also were jointly carried out in positions 1 and 5, which have given rise to compounds with potent anti-neoplasic activity [47,48,49,50].

Here, we present an integrated work including drug synthesis, 3D printing of materials, and biological evaluation of bioactive compounds. Three series of isatin-type biomolecules are presented using supported reagents, either acting as catalysts in CuAAC or Heck reactions [51] as well as isocyanide scavengers (reagents with a very unpleasant odor) in Ugi-type MCRs [52]. The supported reagents are compartmentalized in novel reusable custom porous capsule-type structures, in polypropylene material, using 3D-printing technology. The use of these capsules favors both the dispensing and handling of supported reagents as well as the work up, avoiding filtration stages and recovering the precious reagent for reuse. In addition, we present the preliminary results of the antiproliferative activity of these small series of prepared molecules in tumor cell lines.

## 2. Results and Discussion

The workflow in this study can be summarized in the following points:Preparation of the catalytic systems to be compartmentalized (Figure 1). Synthesis of efficient and robust resin–metal heterogeneous catalysts IRA-120-Cu and IRA-120-Pd (Section 2.1).Computer-aided design of novel capsule prototypes using the editing programs Tinkercad^®^ and Cura^®^ (Section 2.2).Manufacture of PP capsule prototypes using 3D printing (fused deposition modeling) with a semipermeable membrane, enabling the compartmentation of the metal–resin type catalysts of the first point, providing Capsule@IRA-120-Cu(0) and Capsule@IRA-120-Pd(0). In addition, enabling the compartmentation of the isocyanide scavenger Amberlyst-15 to create Capsule@Amberlyst-15 [53]. This supported reagent is a brown-gray granule reticular polystyrene based on ion exchange resins with strong acidic sulfonic groups (Figure 1c). It is used as a strongly acidic heterogeneous acid catalysis and is suitable for nonaqueous catalysis.Study of the chemical stability of the capsule as a resin container, according to the reaction conditions (solvent, base, and temperature), to carry out the CuAAC or Heck reactions as well as the scavenging of isocyanide in Ugi reactions. (Section 2.2).Study of the behavior of the capsule–catalyst pairing. Application to heterogeneous catalysis as CuAAC reactions or PCCCRs (Heck) or as isocyanide scavenger (Ugi). Study of the recyclability of the resins and the capsules. Customization of capsule designs (of different sizes), aiming to carry out reactions at different scales (from milligrams to grams) (Section 2.3 and Section 2.4).

The first efforts were aimed at optimizing appropriate reaction conditions for each isatin series (Figure 2 and Figure 3). Series 1 corresponding to the Ugi reactions (Figure 1) was carried out using a classical procedure, (see Section 2.3) following the methodology of Pineiro et al. but using the Amberlyst-15 resin as a scavenger of excess isocyanide in the work-up process, after the Ugi reactions. The intense and repulsive odor of the isocyanide compounds was described as overpowering and extremely distressing by Hofmann and Gautier [54,55], discouraging many potential contributors to this field. Isocyanides can be extremely unpleasant in odor and toxicity (cyclohexyl isocyanide), which also depends on their degree of volatility. Therefore, the use of scavengers facilitates safe work in the laboratory. The use of PS-TsOH-type resins as an effective isocyanide scavenger was previously reported by our group [56,57]. Series 2 corresponds to three examples of CuAAC to generate triazole-type structures at position 1 of isatin. Series 3 corresponds to examples of Heck-type PCCCRs reactions to generate bioactive structures that have alkenyl functions in position 5 or benzylic substituents in position 1. For the synthesis of series 2 and 3, supported catalysts containing copper or palladium (0) nanoparticles were prepared, respectively.

### 2.1. Synthesis of the Catalytic Materials IRA-120-Cu(0) and IRA-120-Pd(0)

The preparation of the catalytic materials necessary to carry out the syntheses of series 2 and 3, was carried out in a similar way to that described by Silva and coworkers, starting from acidic ion exchange polystyrene resin IRA-120-H [58,59]. The procedure pursued a surface exchange of H^+^ ions for Na^+^ ions after treatment of the IRA-120H resin with NaOH (10%) overnight (Figure 1). Once the new IRA-120Na resin was obtained, it was treated with the appropriate salt (CuSO_4_ in water or Pd(AcO)_2_ in acetone/water) to provide both IRA-120-Cu(II) (green color) and IRA-120-Pd(II) (light brown color) intermediate catalytic materials. A final treatment with NaBH_4_ provided the final resins loaded with copper or palladium nanoparticles, IRA-120-Cu(0) or IRA-120-Pd(0). The final treatment of the intermediate resins produces a drastic change in the color of the resin as a consequence of the change in the oxidation state. The metal nanoparticles (Pd or Cu) remain immobilized in the polyionic environment.

Catalytic resins loaded with copper or palladium species were analyzed by FESEM and EDX experiments. The distribution of both palladium and copper on the 3D-printed spherical bead surfaces were experimentally confirmed. Nanoparticles were found both within the resin and on the surface. The EDX analysis of the samples showed that the loading levels of both species were similar, 21.7% [IRA-Cu(0)] and 22.29% [IRA-Pd(0)]. As can be seen in Figure 1, the dispersion of the metal is more uniform along the surface of IRA-Cu(0) than in IRA-Pd(0).

### 2.2. Fabrication of the Polypropylene Capsules and Compartmentation of Amberlyst 15 and the Catalytic Materials IRA-120-Cu(0) and IRA-120-Pd(0)

We present, here, an innovative, cost effective, simple, and practical porous capsule manufacture for the entrapment of the supported reagents Amberlite-IRA-120-Cu(0) and Amberlite-IRA-120-Pd(0) described in the previous section as well as the commercial Amberlyst-15, to produce Capsule@IRA-120-Cu(0), Capsule@IRA-120-Pd(0), and Capsule@Amberlyst-15, respectively.

#### 2.2.1. Computer-Aided Design (CAD)

These capsules were designed by CAD and synthesized using 3D printing with the aim of miniaturizing and minimizing the work-up operations. The 3D design was performed using the Tinkercad^®^ and Cura^®^ programs. The capsule design itself was thought to be able to carry out the printing and loading processes of the polymeric material in a single stage (see Section 3). After evaluating the possible unicompartmental designs (Figure 2a,e,f,k), a capsule with a bicompartmental design was chosen (Figure 2b,g,j). With this last new design, there is the possibility of introducing a magnetic stirrer into one of the interior compartments and, into the other, the desired resin. Thus, as can be seen in Figure 2, by incorporating a magnetic stirrer inside the capsule, in a different compartment, the resin is not damaged by direct mechanical stress due to contact with the magnetic stirrer. At the same time, it provides the capsule with autonomous movement and flow is very favorable for the exchange of reagents within it. This exchange is favored by certain solvents (by swelling) or by a slight increase in temperature, favoring pore opening of the semipermeable membrane and, therefore, the entry and exit of reactants and products. In any case, the solid catalyst remains inside the capsule during the process. The ultimate aim of these devices is to offer an alternative to the handling of polymeric reagents without the need for orbital stirring devices. 

#### 2.2.2. 3D Printing

Three-dimensional printing is not a new technology but is one of the most revolutionary. One can expect to witness upgrades of the existing design to arrive sooner than with any other technology. Therefore, 3D printing is an effective technology in prototyping processes. Fused deposition modeling (FDM) is one of the most widespread and easily accessible 3D-printing techniques [60]. Its realization begins with the reading of the file that contains the previously chosen design followed by the establishment of the printing conditions selected in advance. The printing process by FDM develops deposition of the selected material and superimposition layer by layer (Figure 2d). Polypropylene (PP) is a semicrystalline thermoplastic polymer with multiple applications thanks to its excellent combination of physicochemical properties, in addition to its relatively low price. Furthermore, PP has a high melting point (160 °C), a high resistance to flexing stress, low water absorption, good electrical and heat resistance, a lightweight, dimensional stability, high impact strength, and a nontoxicity property [61]. The fact that polypropylene contains only carbon and hydrogen atoms in its structure (Figure 1d) implies that this material is only susceptible to being attacked by highly oxidizing agents and halogenated and aromatic solvents. This gives it high chemical resistance. Polypropylene will resist many organic solvents, acids, and alkalis. This material has a low density, which allows the manufacture of very light products. In addition, PP is considered a totally reusable and inert ecological material. Being a semicrystalline material, its crystalline phase provides its structural properties, as well as its rigidity, hardness, and tenacity, while its amorphous phase provides the viscoelastic properties such as its resistance to impacts. All these previously mentioned characteristics make PP a good resistant material to be used in solution chemistry. Nevertheless, a very important aspect is that 3D printing using PP is complicated, because this material has a low adhesion to the glass platforms of the printers. PP tends to stick to itself, but refuses to adhere to other materials, so printing it via FDM is challenging. For this reason, a membrane made of this same material with a minimum pore size was chosen, much less than 0.5 mm (Figure 2c,e), so that it allows the solvent to enter without letting the resin (average particle size of IRA-H: 0.6–0.8 mm, Amberlyst-15: <0.3 mm) pass into the medium reaction during its use in drug synthesis processes. Another noteworthy aspect of our procedure (see Section 3) is that both the selected technique (FDM) and the design of the capsule (bell-shaped or hemisphere) enable the complete fabrication of the device directly and easily in a single step (Figure 2e) while allowing the loading of the polymeric reagent inside (Figure 2f) in a quick process (approx. 20 min). Note that the sealing between the polypropylene membrane of the base (porous) and the capsule under construction occurs instantly, in the first minutes of the printing process. It is through the porous zone that the soluble reagents flow (Figure 2g).

#### 2.2.3. Chemical and Mechanical Resistance Tests under SPOS Conditions

During the preliminary optimization stage, a series of capsule resistance tests were carried out against various solvent and temperature conditions. The great chemical and mechanical stability of PP [62] at high temperatures in a solution of methanol, water, and DMF was confirmed (Figure 2k). However, when solvents such as toluene or DCM were used, the capsule was damaged, so these devices cannot be used in chemical reactions that contain these solvents when working at high temperatures.

### 2.3. Preparation of Series 1 (Compounds ***11***–***22***): Ugi Reaction

Ugi compounds ([Fig pharmaceuticals-16-00310-ch001]) were synthesized using similar conditions to those reported by Burke, Pineiro et al. [43]. These derivatives with unsubstituted NH at the isatin ring and halogen or alkyl substituents at position 5 reached cytotoxicity efficacies at the range of the most active refence compound, paclitaxel, against HBL-100 and HeLa cell lines. Specifically, this work aims to also report the activity of new diversely substituted isatins derived from the most potent structures reported previously [43] as cytotoxic agents (Figure 2). Particularly, compound **12** showed potent antiproliferative activity in HBL-100 and HeLa lines, thus being considered as the initial hit. Taking these results into account, which seem to suggest the importance of certain substituents at position 3 and 5 of the isatin ring, we selected a small library of promising target compounds to be tested in the murine 3T3-Balb fibroblast and human tumoral cervical HeLa cell lines containing these building blocks and, at the same time, verifying the effectiveness of the Capsule@Amberlyst-15-type resin (PS-TsOH) in the work-up processes for the elimination of excess isocyanide. The reactions proceeded satisfactorily in the presence of a Lewis acid-type catalyst (InCl_3_) in methanol. In these transformations, we verified that the pKa of the acid component is key to the success of the reaction. Thus, reactions using trifluoroacetic acid work better and faster than those with chloroacetic acid. The reaction times and the yields are shown in Table 1. The expected Ugi-isatines **11**–**22** were also completely characterized by NMR experiments and compound **17** by single crystal X-ray analysis. Images of the ORTEP diagram corresponding to the crystalline structure are shown in Figure 3.

### 2.4. Capsule@Amberlyst-15 Behavior as Scavenger of Isocyanides, Post-Ugi Isocyanide Removal, Recycling of the Capsule after Scavenging

Scavenging process: The behavior of the Capsule@Amberlyst-15 capsule–resin pairing was evaluated during the work-up process for the removal of excess isocyanide. Once the final product was formed, monitored by TLC, the scavenging process was carried out by directly immersing a semiporous polypropylene capsule containing the scavenger in the reaction medium, maintaining the mixture under magnetic stirring. Two equivalents of the supported p-TsOH led to complete elimination of the reactants under mild conditions. The process is relatively fast (30 min–1 h). After that interval, a complete absence of isocyanide odor is observed. After the process is complete, the capsule is removed from the flask with tweezers and washed with methanol. 

Recycling process: The reactivation of the Capsule@Amberlyst-15 was carried out following a protocol described previously by our group [56]. After appropriate washing protocol (Figure 4d), the capsule containing the recovered sulfonate salt (formed after the scavenging process) was incubated in a 30% solution of trifluoroacetic acid in dichloromethane and submitted to orbital stirring for 1 h at room temperature (Figure 4). Once appropriately washed and dried under vacuum, the recovered immobilized capsule containing the p-toluenesulfonic acid supported reagent can be reused at least five times in new scavenging experiments for the same isocyanide without dramatic loss of effectiveness. Additionally, no apparent fracture or damage was detected in the capsule after treatment with dilute TFA.

### 2.5. Preparation of Series 2 (Compounds ***25***–***27***) and 3 (Compounds ***30***–***33***)

The synthetic route to access new isatin-derived series 2 and 3 is shown in Figure 3. The starting isatins were alkylated with propargyl bromide, benzyl bromide, or 4-iodo-benzyl bromide to provide the key intermediates **23**, **24**, **28**, and **29**, respectively [47,63]. The CuAAC products, **25**–**27**, were obtained by treating the starting alkynyl isatins with organic azides (benzyl azide or ethyl 2-azidoacetate) in the presence of a capsule containing the capsule@IRA-120-Cu(0), without the presence of other additives. The choice of the solvent was key: the starting isatins are not very soluble in EtOH or tBuOH at room temperature, although these solvents are frequently used in CuAAC [64] and compatible with capsule stability. Acetonitrile was optimal to obtain compounds **25**–**27** with good yields and in short reaction times, which demonstrates a correct flow of reagents within the capsule. In this sense, a bicompartmental (internal magnetic stirrer) capsule shakes more vigorously than a unicompartmental one, accelerating the transformations inside the capsule.

The protocol for the synthesis of series 3 consisted of the preparation of four compounds derived from the dioxoindolinyl acrylate moiety. Three of them (**30**, **32**, **33**) were not previously described by Peng Yu et al. [47]. Thus, for the synthesis of compounds **30**–**33**, the Heck reaction for the alkenylation of halogenated positions assisted by the IRA-120-Pd(0) catalyst, loaded in a bicompartmental capsule (Figure 5) was used. The choice of the appropriate solvent (DMF/H_2_O), base (sodium acetate), and temperature were crucial, since very high temperatures may damage both the capsule and the final isatin product itself, generating reaction byproducts.

In the case of the synthesis of compound **30**, starting from N-deprotected isatin **2**, in addition to the Heck alkenylation in position 5, a 1,4-Aza-Michael addition occurs, promoted by deprotonation at N-1. For the synthesis of **31**, **32**, and **33**, precursors **28** and **29** were prepared, containing benzylic residues in position 1. The presence of a second iodine atom in the molecule, in para position of the benzylic ring, allows both obtaining the monoalkenylation product **32** and/or the double alkenylation product **33** (Table 2). The reactivity towards the iodo of C-5 to create compound **32** can be enhanced by conducting the reaction at moderate temperature (70 °C). On the contrary, a higher temperature (90 °C) and an excess of acrylate lead to a higher proportion of the bialkenylated compound **33**. Their structures were confirmed by ^1^H NMR and ^13^C NMR as well as ESI–MS.

### 2.6. Capsule Behavior and Reuse, Leaching Studies

These capsule devices have excellent resistance to the conditions of temperature (70–90 °C), reagents, and solvents required for the Heck reactions, as well as for CuAAC. In addition, the use of capsules facilitates the work-up process through extraction by tweezers. As can be seen in Figure 5, the shape and size of the 3D-printed prototypes are adapted to the dimensions of the desired flask. As can be seen in Figure 5c, it is possible to manufacture devices of different sizes and shapes. The bicompartmental capsules used in this work (25 × 17.7 × 19.6 mm) offer an approximate loading capacity of 900 mg of the three resins. Each capsule was reused in at least 5 new reactions without apparent loss of efficacy. 

Interestingly, the polymer reagents contained in the capsules were inspected after 5 reaction cycles. The images taken under the microscope did not show significant changes in terms of the form of the supported reagents nor in the nanoparticle content. IRA-Pd(0) remained black while IRA-Cu(0) experienced a slight darkening of its copper color, probably as a consequence of a superficial oxidation process of copper. However, the recovery of the “zero” oxidation state for copper (clearly evidenced by the instantaneous color change) can easily be carried out by treatment with aqueous NaBH_4_ (5 equiv.) for 1 h, at room temperature.

### 2.7. Biological Activity, Evaluation of Antiproliferative Activity

The cytotoxic activity of these new synthesized isatin derivatives was evaluated by means of the CCK8 antiproliferation assay in vitro in two different cell lines: cancerous cervical HeLa and murine 3T3 Balb fibroblast cells. The latter cell line is typically used to analyze the potential toxicity of (nano)materials thanks to their sensitivity. Figure 6 shows that isatins display a concentration-dependent cytotoxic profile. It can be observed that fibroblast cells were more sensitive to the presence of isatin derivatives than the cancerous cells in general HeLa ones, probably related to the enhanced replication rate of the tumoral cells. Analysis of the experimental data shown in Figure 6 allowed the determination of IC_50_ values (see Table 3 and Table 4). It is possible to discern two highly active molecules, compounds **26** and **33**, with high specificity for tumoral cells and rather low half-inhibitory concentration values, IC_50_ of 2.5 and 1.0 μM, respectively, and which is also much lower than those of Pt-based chemodrugs such as cisplatin (IC_50_ = 15 μM) and carboplatin (IC_50_ = 1.0 mM) currently used as first line treatment in clinics of primary cervical tumors. [65,66] Compounds **26** and **33** are, therefore, new promising hits for further development.

Other drugs such as paclitaxel (IC_50_ = 21 μM) and topotecan (IC_50_ = 26 μM) [66,67] are used for the therapeutics of recurrent and metastatic cervical cancer. Moreover, there are another three compounds, **12**, **30**, and **31**, which despite not showing fully specific toxic activity for cancerous cells as occurred for many other potential chemotherapeutic compounds, they also have reduced IC_50_ values of 8.3, 9.3, and 6.8 μM, respectively. Nevertheless, further studies are required to test the present synthesized isatin derivatives in other cancerous cells lines as toxicity profiles are known to depend on cell phenotype (see, for example, compounds **11**, **14**, and **22**), in addition to identifying the potential mechanism/s involved in cytotoxicity and subsequent cell death.

The data commented above, referring to the antiproliferative activity, allow us to draw some preliminary conclusions regarding the structure–activity relationships in these series. Thus, in series 1 (Ugi compounds), the maintenance of the chloroacetamide residue in position 3 seems to be important, as well as the presence of halogens (Br, I, Cl) in position 5 of isatin. With these characteristics, potent inhibition is achieved in at least one of the two cell lines studied, particularly for compound **22** (IC_50_ = 0.12 μM in Balb Mur cell line) and **12** (IC_50_: 1.3 μM in Balb and 8.3 μM in HeLa). Regarding the substituent from isocyanide (tert-butyl or cyclohexyl) from the Ugi reaction, both lyphophilic residues can provide good levels of antiproliferative activity, so this seems to be a less restrictive region.

Regarding series 2 and 3, we can conclude that the presence of a benzylic function on nitrogen is important for the activity in HeLa. In general, the aromatic moieties (benzene or 1,2,3-triazole) are conformationally flexible (see compounds **26** and **33**). The presence of an acrylate substituent in position 5 significantly improves potency (see compounds **31** and **33**). Substitution of the benzyl group by a short aliphatic chain containing an ester moiety does not improve activity (**31**). However, the presence of a 1,2,3-triazole ring in the place of benzene also seems to be very effective (**26**) when the halogen is kept at 5. New studies will be necessary to evaluate the best combination of substituents, both on the benzylic (or triazol) function as on position 5 of the isatin moiety, where both halogens (iodo) and the methyl acrylate-type residues are effective.

## 3. Materials and Methods

### 3.1. Chemistry: General Procedures for the Syntheses of the Catalytic Materials, Programs, Reagents, and Materials

Images of Pd nanoparticles dispersed in the polymer IRA-120 were acquired using a Gemini-500 field-emission scanning electron microscope (FESEM) operating at 20 kV using a back-scattering AsB detector with a size resolution of ±0.5 nm. Selected images were analyzed by counting more than 100 particles using ImageJ software. The surface morphology and microstructure of the samples were characterized using a scanning electron microscope (SEM, JEOL 6400, JEOL Corporation, Japan) and a stereomicroscope (OlympusSZX12, Olympus, Japan). The surface elemental analysis of sintered samples was measured using an energy dispersive X-ray spectrometer (EDS, AZTEC/Xact, Oxford, UK).

Synthesis of IRA-120-Cu(0): Immobilization of copper (0) species on IRA-120H (1.8 mmol/g loading), support: Kimble vials in a PLS (6 × 4) organic synthesizer were used to load IRA-120H. First, IR-120 (1 g) was treated with 50 mL of NaOH 10% and stirred overnight at rt. The resulting light brown solution was filtered. The brown solid support IRA-120Na (0.5 g) was dried and then treated with a solution of CuSO_4_ pentahydrate (25 mg, 0.1 mmol) in 10 mL of H_2_O, and the suspension was vigorously stirred under orbital stirring at room temperature for 24 h. After this period, the blue color of the initial copper solution disappeared. The resulting green IRA-120-Cu(II) was filtered, washed with water, and dried under vacuum for 1 h at room temperature. The catalysts on the solid resin support were then filtered through a filter plate, washed, and transferred to a beaker with 50 mL of water. To this mixture, a fresh solution of 5 mL of NaBH_4_ (1 mol/L) were added. The reduction was readily evidenced by the color change (copper color). IRA-120-Pd(0) catalytic system was prepared using the same procedure above but using Pd(AcO)_2_ (20 mg) as metallic salt. 

### 3.2. Manufacture of Polypropylene Capsules by 3D Printing

Tinkercad and CURA programs were used for the construction of the virtual 3D capsule. An Ultimaker 2+ extended 3D printer was used for the construction of the capsule, using the fused deposition modeling technique. A filament of polypropylene (2.85 mm diameter) was used as a 3D-printing material for the capsule. A piece of porous polypropylene membrane sheet was placed on the base and fixed to the base with a polypropylene film. The base conditions for the 3D printing of polypropylene (PP) capsules were the following. The platform was maintained at 40 °C. Nozzle diameter and temperature: 0.6 mm, 165 °C. The 3D-printed structure was sealed at the bottom by melting the borders of the polypropylene membrane during 3D printing. Once the printing of the capsule had begun (when the process reached 50% progress and the capsule walls had already formed, Appendix A), the printing was paused briefly to fill the capsule with the IRA-120-Cu or Pd resin. Once the capsule was filled with the polymer reagent (and the magnetic stirrer in the case of bicompartimental capsules), 3D printing was resumed. To avoid melting or swelling of the thin polypropylene membrane located in the base, the temperature of the platform was maintained at 40 °C. The printing of the capsule was completed in 23 min (for dimensions: 25 × 17 × 19 mm).

### 3.3. Chemistry, General Procedures for the Syntheses of the Series 1, 2, and 3

Reagents and materials: Kimble vials in a PLS (6 × 4) organic synthesizer were used to perform the functionalization of IRA support with Pd and Cu species. Polystyrene-supported IRA-120H (1.8 mmol/g loading) and Amberlyst-15 (4.7 mmol/g loading) were purchased from Fluka. The rest of the reagents, methyl acrylate, isatins, InCl_3_, were provided by Sigma-Aldrich. All reactions were monitored by TLC with 2.5 mm Merck silica gel GF 254 strips. The final purified compounds showed a single spot. Detection of compounds was performed by UV light and/or iodine vapor. Purification of isolated products was carried out by preparative TLC using silica gel plates. Characterization of the synthesized compounds was performed using spectroscopic and analytical data. The NMR spectra were recorded on Bruker AM 400 MHz (^1^H) and 75 MHz (^13^C) and XM500 spectrometers. Chemical shifts are given as δ values against tetramethylsilane as the internal standard. J values are given in hertz. Proton and carbon nuclear magnetic resonance spectra were recorded in CDCl_3_ or DMSO. Melting points were determined on a Gallenkamp apparatus and are uncorrected. Mass spectra were obtained on a Varian MAT-711 instrument. Mass spectra (ESI-MS) were obtained on an autospec micromass spectrometer. Crystalographic data of compound **17**: the data was measured with the Bruker D8 VENTURE PHOTON-III diffractometer using Mo radiation and at a temperature of 100K. Computing details of data collection: Bruker APEX4 software; cell refinement: SAINT V8.40B (Bruker AXS LLC, 2019); data reduction: SAINT V8.40B (Bruker AXS LLC, 2019); program(s) used to solve structure: SHELXT 2018/2 (Sheldrick, 2015); program(s) used to refine structure: SHELXL2019/1 (Sheldrick, 2019); molecular graphics: ORTEP 2014.1 (Farrugia, 2012); software used to prepare material for publication: IUCr Journals pCIF (see Appendix A).

### 3.4. General Procedure for the Syntheses of Series 1 (Compounds ***11***–***22***)

The chemical library of Ugi-compounds was prepared by dissolving the corresponding isatin (**1**–**5**) (1 mmol), InCl_3_ (10%), 100 mg of molecular sieves (3Å), n-butylamine (1.5 mmol), the acid component (1.5 mmol), and the isocyanide (1.5 mmol) in MeOH. After 48–72 h, CH_2_Cl_2_ (10 mL) and the bicompatmental Capsule@Amberlyst-15 (containing 900 mg of resin) were added. The mixture was stirred at rt for 1 h. Then, the capsule was extracted from the mixture, washed with MeOH and CH_2_Cl_2_ for reuse in further reactions after reactivation. After solvent removal under reduced pressure, the crude product was crystallized from ethanol or purified by preparative TLC (CH_2_Cl_2_/MeOH). Reactivation of bicompatmental Capsule@IRA-Amberlyst-15: The reagent-containing capsule was treated with 30 mL of a TFA/CH_2_Cl_2_ (30%) solution for 3 h while stirring and at room temperature.

Compound N-(tert-butyl)-3-(N-butyl-2-chloroacetamido)-1-methyl-2-oxoindoline-3-carboxamide (**11**) was obtained as a white solid (316.8 mg, 65% yield). m.p. = 144.1–145.0 °C. ^1^H NMR (CDCl_3_, 400 MHz): δ 7.48 (d, *J* = 7.0 Hz, 1H), 7.32 (td, *J* = 7.7, 1.1, 1H), 7.18 (br, 1H), 7.10 (td, *J* = 7.6, 0.7 Hz, 1H), 6.84 (d, *J* = 7.8 Hz, 1H), 4.09 (dd, *J* = 13.0 Hz, 2H), 3.69–3.3.61 (m, 1H), 3.59–3.50 (m, 1H), 3.26 (s, 3H), 1.94–1.84 (m, 1H), 1.74–1.63 (m, 1H), 1.36–1.30 (m, 11H), 0.95 (t, *J* = 7.4 Hz, 3H). MS (ESI) [M-H]: 378 [43].

Compound N-(tert-butyl)-3-(N-butyl-2-chloroacetamido)-5-iodo-2-oxoindoline-3-carboxamide (**12**) was obtained as a yellow solid (102.9 mg, 28% yield). m.p. = 108.5–109.7 °C. ^1^H NMR (CDCl_3_, 400 MHz): δ 8.11 (br s, 1H), 7.68 (d, *J* = 1.6 Hz, 1H), 7.54 (dd, *J* = 8.2, 1.7 Hz, 1H), 7.17 (br s, 1H), 6.60 (d, *J* = 8.2 Hz, 1H), 4.17 (d, *J* = 13.2 Hz, 1H), 4.10 (d, *J* = 13.1 Hz, 1H), 3.70–3.62 (m, 1H), 3.58–3.50 (m, 1H), 1.97–1.86 (m, 1H), 1.78–1.67 (m, 1H), 1.37–1.30 (m, 11H), 0.99 (t, *J* = 7.3 Hz, 3H). MS (ESI) [M-H]: 504 [43].

Compound N-(tert-butyl)-3-(N-butyl-2-chloroacetamido)-5-nitro-2-oxoindoline-3-carboxamide (**13**) was obtained as a white solid (230.7 mg, 52% yield). m.p. = 179.5–179.8 °C. ^1^H NMR (CDCl_3_, 400 MHz): δ 8.67 (br s, 1H), 8.29 (d, *J* = 2.2 Hz, 1H), 8.21 (dd, *J* = 8.6, 2.3 Hz, 1H), 7.16 (br s, 1H), 6.90 (d, *J* = 8.6 Hz, 1H), 4.24 (d, *J* = 13.3 Hz, 1H), 4.12 (d, *J* = 13.3 Hz, 1H), 3.76–3.68 (m, 1H), 3.66–3.58 (m, 1H), 2.10–1.99 (m, 1H), 1.88–1.76 (m, 1H), 1.42 (q, *J* = 7.4 Hz, 2H), 1.35 (s, 9H), 1.04 (t, *J* = 7.3 Hz, 3H). MS (ESI) [M-H]: 423 [43].

Compound N-(tert-butyl)-3-(N-butyl-2-chloroacetamido)-5,7-dichloro-2-oxoindoline-3-carboxamide (**14**) was obtained as a white-yellow solid (30% yield). m.p.= 125–126 °C. ^1^H NMR (CDCl_3_, 400 MHz): δ 8.94 (br, 1H, NH), 7.43 (s, 1H), 7.38 (s, 1H), 4.28 (dd, *J* = 9.5, 1.2 Hz, 2H), 3.80–3.64 (m, 2H), 2.05 (m, 1H), 188 (m, 1H), 1.59–1.40 (m, 12H), 1.15–105 (m, 4H). ^13^C NMR (CDCl_3_, 75.4 MHz MHz): δ 174.6, 167.1, 159.9, 137.5, 128.9, 128.7, 115.7, 109.2, 73.1, 52.5, 47.1, 41.4, 33.8, 31.2, 28.1, 19.9, 13.5. MS (ESI) [M-H]: 446. 

Compound 5-bromo-3-(*N*-butyl-2,2,2-trifluoroacetamido)-*N*-cyclohexyl-2-oxoindoline-3-carboxamide (**15**) was obtained as a white-yellow solid (58% yield). m.p. = 209–210 °C. ^1^H NMR (CDCl_3_, 400 MHz): δ 8.91 (br, 1H), 7.59 (s, 1H), 7.37 (dd, *J* = 8.3, 2.1 Hz, 1H), 6.96 (d, *J* = 7.8 Hz, 1H), 6.69 (d, *J* = 8.2 Hz, 1H), 3.86–3.55 (m, 3H), 1.94–1.14 (m, 14H), 0.92 (t, *J* = 7.4 Hz, 3H). ^13^C NMR (CDCl_3_, 75.4 MHz MHz): δ 173.8, 160.1, 140.1, 132.9, 129.3, 127.2, 117.3, 116.1, 114.5, 112.0, 72.1, 49.7, 46.9, 33.3, 32.1, 32.1, 25.3, 24.5, 24.4, 19.8, 13.3. MS (ESI) [M-H]: 503. 

Compound 5-Bromo-N-(tert-butyl)-3-(N-butyl-2-chloroacetamido)-2-oxoindoline-3-carboxamide (**16**) was obtained as a pale-yellow solid (85.6 mg, 21% yield). m.p. = 180–182 °C. ^1^H NMR (CDCl_3_, 400 MHz): δ 8.14 (br s, 1H), 7.52 (d, *J* = 1.8 Hz, 1H), 7.35 (dd, *J* = 8.3, 2.0 Hz, 1H), 7.17 (br s, 1H), 6.69 (d, *J* = 8.3 Hz, 1H), 4.18 (d, *J* = 13.1 Hz, 1H), 4.10 (d, *J* = 13.1 Hz, 1H), 3.70–3.62 (m, 1H), 3.59–3.50 (m, 1H), 1.99–1.88 (m, 1H), 1.79–1.68 (m, 1H), 1.39–1.34 (m, 11H), 0.99 (t, *J* =7.3 Hz, 3H). MS (ESI) [M-H]: 457 [43].

Compound *N*-(*tert*-butyl)-3-(*N*-butyl-2,2,2-trifluoroacetamido)-5-iodo-2-oxoindoline-3-carboxamide (**17**) was obtained as a white solid (60% yield). m.p.= 204–205 °C. ^1^H NMR (CDCl_3_, 400 MHz): δ 8.50 (br s, 1H), 7.72 (s, 1H), 7.58 (dd, *J* = 8.2, 1.7 Hz, 1H), 7.01 (s, 1H), 6.60 (d, *J* = 8.2 Hz, 1H), 3.81 (m, 1H), 3.61 (m, 1H), 1.94 (m, 1H), 1.68 (m, 1H), 1.32 (m, 11H), 0.95 (t, *J* = 7.4 Hz, 3H). ^13^C NMR (CDCl_3_, 75.4 MHz MHz): δ 174.0, 159.5, 157.7, 140.7, 138.7, 135.1, 127.5, 117.3, 112.4, 86.1, 72.5, 52.6, 46.9, 33.5, 28.1, 19.9, 13.4. MS (ESI) [M-H]: 526. 

Compound 3-(*N*-butyl-2,2,2-trifluoroacetamido)-*N*-cyclohexyl-5-iodo-2-oxoindoline-3-carboxamide (**18**) was obtained as a white-yellow solid (55% yield). m.p.= 218–220 °C. ^1^H NMR (CDCl_3_, 400 MHz): δ 8.70 (s, 1H), 7.75 (s, 1H), 7.57 (dd, *J* = 8.2, 1.7 Hz, 1H), 6.96 (d, *J* = 7.8 Hz, 1H), 6.58 (d, *J* = 8.2 Hz, 1H), 3.86–3.55 (m, 3H), 1.95–1.14 (m, 14H), 0.92 (t, *J* = 7.4 Hz, 3H). ^13^C NMR (CDCl_3_, 75.4 MHz MHz): δ 173.6, 160.1, 157.4, 140.7, 138.8, 134.9, 127.5, 117.5, 117.3, 112.4, 86.1, 71.9, 49.7, 46.8, 33.3, 32.1, 32.1, 25.3, 24.5, 24.4, 19.8, 13.4. MS (ESI) [M-H]: 550.

Compound 5-bromo-*N*-(*tert*-butyl)-3-(*N*-butyl-2,2,2-trifluoroacetamido)-2-oxoindoline-3-carboxamide (**19**) was obtained as a white solid (58% yield). m.p.= 198–200 °C. ^1^H NMR (CDCl_3_, 400 MHz): δ 8.37 (br s, 1H), 7.57 (s, 1H), 7.39 (dd, *J* = 8.3, 2.0 Hz, 1H), 7.01 (s, 1H), 6.70 (d, *J* = 8.2 Hz, 1H), 3.82 (m, 1H), 3.62 (m, 1H), 1.96 (m, 1H), 1.69 (m, 1H), 1.32 (m, 11H), 0.95 (t, *J* = 7.4 Hz, 3H). ^13^C NMR (CDCl_3_, 75.4 MHz MHz): δ 174.1, 159.5, (151.0), 140.1, 132.8, 129.5, 127.2, 116.1, 112.0, 86.1, 72.7, 52.6, 46.9, 33.5, 28.1, 19.8, 13.4. MS (ESI) [M-H]: 477. 

Compound *N*-(*tert*-butyl)-3-(*N*-butyl-2,2,2-trifluoroacetamido)-5-nitro-2-oxoindoline-3-carboxamide (**20**) was obtained as a white solid (55% yield). m.p.= 178–180 °C. ^1^H NMR (CDCl_3_, 400 MHz): δ 8.89 (br, 1H), 8.33 (d, *J* = 2.3 Hz, 1H), 8.23 (dd, *J* = 8.7, 2.3 Hz, 1H), 7.06 (s, 1H), 6.94 (d, *J* = 8.6 Hz, 1H), 3.87 (m, 1H), 3.67 (m, 1H), 2.09 (m, 1H), 1.79 (m, 1H), 1.35 (m, 11H), 1.00 (t, *J* = 7.4 Hz, 3H). ^13^C NMR (CDCl_3_, 75.4 MHz MHz): δ 174.9, 158.5, 146.5, 144.2, 126.6, 126.1, 122.5, 117.2, 114.34, 110.3, 72.3, 523.0, 47.1, 33.7, 28.1, 19.9, 13.4. MS (ESI) [M-H]: 443. 

Compound 3-(*N*-butyl-2,2,2-trifluoroacetamido)-*N*-cyclohexyl-5-nitro-2-oxoindoline-3-carboxamide (**21**) was obtained as a pale-yellow solid (57% yield). m.p.= 200–202 °C. ^1^H NMR (CDCl_3_, 400 MHz): δ 8.98–8.78 (br, 1H), 8.36 (s, 1H), 8.22 (dd, *J* = 8.7, 2.3 Hz, 1H), 7.05 (d, *J* = 7.8 Hz, 1H), 6.90 (d, *J* = 8.6 Hz, 1H), 3.95–3.61 (m, 3H), 1.98–1.18 (m, 14H), 0.98 (t, *J* = 7.4 Hz, 3H). ^13^C NMR (CDCl_3_, 75.4 MHz MHz): δ 174.6, 158.9, 146.6, 144.1, 144.1, 126.6, 126.1, 122.3, 117.2, 114.3, 110.3, 71.7, 50.0, 47.1, 33.5, 32.1, 25.2, 24.5, 24.4, 19.9, 13.4. MS (ESI) [M-H]: 469. 

Compound 5-bromo-3-(*N*-butyl-2-chloroacetamido)-*N*-cyclohexyl-2-oxoindoline-3-carboxamide (**22**) was obtained as a beige solid (42% yield). m.p.= 190.2–190.4 °C. ^1^H NMR (CDCl_3_, 300 MHz): δ 8.65 (s, 1H), 7.54 (s, 1H), 7.33 (dd, *J* = 8.3, 2.0 Hz, 1H), 7.16 (d, *J* = 7.8 Hz, 1H), 6.66 (d, *J* = 8.3 Hz, 1H), 4.15 (q, *J* = 13.2 Hz, 2H), 3.73–3.46 (m, 3H), 1.95–1.15 (14H), 0.97 (t, *J* = 7.3 Hz, 3H). ^13^C NMR (CDCl_3_, 75.4 MHz MHz): δ 174.8, 167.1, 160.9, 140.2, 132.3, 129.1, 128.2, 115.7, 111.9, 71.8, 49.5, 47.2, 41.62, 33.6, 32.2, 25.3, 24.5, 24.5, 20.1, 13.5. MS (ESI) [M-H]: 483. 

### 3.5. General Procedures for the Syntheses of the Intermediates for Series 2 and 3

To a solution of isatins **1** and **2** (1 mmol) was added anhydrous K_2_CO_3_ (1.5 mmol, 207 mg) in anhydrous acetonitrile (10 mL) in a microwave vessel. Propargyl bromide (1.5 mmol, 0.167 mL of solution 80 wt% in toluene) was then added dropwise into the suspension mixture. The reaction mixture was heated under reflux while stirring at 50 °C for 12 h. When the reaction was accomplished (monitored by TLC.), the solvent was evaporated completely under reduced pressure at 50 °C. After the reaction was completed, the mixture was cooled to room temperature, and water was added to the mixture to dissolve inorganic salts (K_2_CO_3_ and KBr) and extracted with ethyl acetate and dried with anhydrous sodium sulfate. The solvent was removed under reduced pressure. The residue was crystallized from ethanol to afford the title substituted N-propargyl isatins **23** or **24** [63].

Compound **23** was obtained as an orange solid (80% yield). m.p.= 156–158 °C. ^1^H NMR (CDCl_3_, 400 MHz): δ 7.65 (m, 2H), 7.14 (m, 2H), 4.53 (s, 2H), 2.31 (s, 1H). ^13^C NMR (CDCl_3_, 75.4 MHz MHz): δ 182.5, 157.1, 149.6, 138.4, 125.4, 124.2, 117.7, 111.1, 75.6, 73.3, 29.4. MS (ESI) [M + H]: 186.0. 

Compound **24** was obtained as an orange solid (85% yield). m.p.= 110–112 °C. ^1^H NMR (CDCl_3_, 400 MHz): δ 7.92 (m, 2H), 6.95 (m, 1H), 4.52 (s, 2H), 2.32 (s, 1H). ^13^C NMR (CDCl_3_, 75.4 MHz MHz): δ 181.1, 156.1, 148.8, 146.4, 133.9, 119.2, 113.2, 86.6, 75.2, 73.7, 29.5. MS (ESI) [M + H]: 311.90.

To a solution of isatin **2** (1 mmol) was added anhydrous K_2_CO_3_ (1.5 mmol, 207 mg) in anhydrous acetonitrile (10 mL) in a microwave vessel. Benzyl bromide or 4-iodo-benzyl bromide (1.5 mmol) was then added dropwise into the suspension mixture. The reaction mixture was heated under reflux while stirring at 50 °C for 12 h. When the reaction was accomplished (monitored by TLC.), the solvent was evaporated completely under reduced pressure at 50 °C. After the reaction was completed, the mixture was cooled to room temperature, and water was added to the mixture to dissolve inorganic salts (K_2_CO_3_ and KBr) and extracted with ethyl acetate and dried with anhydrous sodium sulfate. The solvent was removed under reduced pressure. The residue was crystallized from ethanol to afford the title substituted N-benzyl isatins **28** or **29**.

Compound **28** was obtained as an orange solid (90% yield). m.p.= 200–201 °C. ^1^H NMR (CDCl_3_, 400 MHz): δ 7.86 (m, 2H), 7.53 (s, 1H), 7.38–7.18 (m, 6H), 5.48 (s, 2H), 4.97 (s, 2H). ^13^C NMR (CDCl_3_, 75.4 MHz MHz): δ 181.7, 157.2, 149.6, 146.4, 138.3, 134.0, 133.7, 129.2, 119.2, 112.9, 93.9, 86.4, 43.6. MS (ESI) [M + H]: 321.33.

Compound 1-benzyl-5-iodoindoline-2,3-dione (**29**) was obtained as an orange solid (90% yield). m.p.= 198–200 °C. ^1^H NMR (CDCl_3_, 400 MHz): δ 7.89 (d, *J* = 1.8 Hz, 1H), 7.77 (d, *J* = 8.3 Hz, 1H), 7.40–7.24 (m, 5H), 6.57 (d, *J* = 8.3 Hz, 1H), 4.92 (s, 2H). ^13^C NMR (CDCl_3_, 75.4 MHz MHz): δ 181.9, 157.2, 159.0, 146.3, 134.0, 133.8, 129.1, 128.3, 127.4, 119.2, 113.1, 86.2, 44.1. MS (ESI) [M + H]: 363. 

### 3.6. General Procedures for the Syntheses of the Serie 2 (CuAAC Reaction)

In a round-bottomed flask, alkynyl isatin **23** (400 mg, 1 mmol) was dissolved in acetonitrile (10 mL). Then, benzyl azide (1.1 mmol) and Capsule@IRA-120-Cu(0) (containing 100 mg of resin, containing 21% Cu) were added. The reaction was left stirred at room temperature for 10 h. The reaction was monitored by TLC. Once the complete transformation of the starting product was verified, the capsule was removed from the reaction medium with tweezers and washed with acetonitrile for reuse. The organic solution used for washing the capsule was combined with the rest of the reaction mixture and evaporated to dryness in a rotary evaporator. The resulting solid was recrystallized from iPrOH to create **25**.

Compound **25** was obtained as an orange solid (95% yield). m.p.= 175–177 °C. ^1^H NMR (CDCl_3_, 400 MHz): δ 7.51–6.99 (m, 10H), 5.41 (s, 2H), 4.91 (s, 2H). ^13^C NMR (CDCl_3_, 75.4 MHz MHz): δ 183.1, 158.1, 150.3, 142.0, 138.6, 134.0, 129.2, 129.0, 128.3, 125.3, 124.0, 123.2, 117.6, 111.6, 54.7, 35.4. MS (ESI) [M + H]: 319.1. 

Compound **26** was obtained as an orange solid (90% yield). m.p.= 183–185 °C. ^1^H NMR (CDCl_3_, 400 MHz): δ 7.86 (m, 2H), 7.53 (s, 1H), 7.38–7.18 (m, 6H), 5.48 (s, 2H), 4.97 (s, 2H). ^13^C NMR (CDCl_3_, 75.4 MHz MHz): δ 181.7, 156.9, 149.5, 146.7, 142.0, 133.8, 133.7, 129.3, 129.1, 128.3, 123.0, 119.0, 113.8, 86.4, 54.6, 35.3. MS (ESI) [M + H]: 311.08.

Compound **27** was obtained as a red solid (93% yield). m.p.= 158–159 °C. ^1^H NMR (CDCl_3_, 400 MHz): δ 7.86–7.80 (m, 3H), 7.12 (dd, *J* = 8.3, 0.6 Hz, 1H), 5.13 (s, 2H), 5.00 (s, 2H), 4.23 (q, *J* = 7.1 Hz, 2H), 1.26 (t, *J* = 7.1 Hz, 3H). ^13^C NMR (CDCl_3_, 75.4 MHz MHz): δ 181.7, 165.9, 156.9, 149.5, 146.6, 133.7, 133.6, 124.5, 119.0, 113.7, 86.4, 62.5, 50.9, 35.4, 14.0. MS (ESI) [M + H]: 441. 

### 3.7. Syntheses of the Series 3 (Heck Reaction)

In a round bottom flask, 400 mg (1 mmol) of 5-iodoisatin **29** was dissolved in DMF (10 mL). To this solution was added another aqueous solution of sodium acetate (3 mmol in 1.5 mL of water), methyl acrylate (1,5 mmol), and Capsule@IRA-120-Pd(0) (100 mg of resin, containing 22% of Pd). The reaction was kept stirring at 90 °C for 12 h. The reaction was monitored by TLC until complete consumption of the starting product. Once the reaction was finished, the capsule was extracted from the reaction medium and washed (MeOH, water, AcOEt) for reuse. The reaction mixture was diluted with AcOEt (30 mL) and washed with water (10 mL), and the solvent was evaporated in a rotavapor at 50 °C, and the residue was purified on preparative TLC to create compound **31**. 

Compound 1-benzyl-5-iodoindoline-2,3-dione **31** was obtained as a red solid (88% yield). m.p.= 164–165 °C. ^1^H NMR (CDCl_3_, 400 MHz): δ 7.70 (d, *J* = 1.8 Hz, 1H), 7.57–7.46 (m, 2H), 7.30–7.19 (m, 5H), 6.74 (d, *J* = 8.2 Hz, 1H), 6.27 (d, *J* = 16.0 Hz, 1H), 4.88 (s, 2H), 3.72 (s, 3H). ^13^C NMR (CDCl_3_, 75.4 MHz MHz): δ 182.6, 166.9, 158.1, 151.5, 142.4, 138.0, 134.1, 130.5, 129.1, 128.3, 127.4, 124.1, 118.2, 119.0, 111.4, 51.8, 44.2. MS (ESI) [M + H]: 322.3.

In a round bottom flask, 400 mg (1 mmol) of 5-iodoisatin **29** was dissolved in DMF (10 mL). To this solution was added another aqueous solution of sodium acetate (3 mmol in 1.5 mL of water), methyl acrylate (1,2 mmol), and Capsule@IRA-120-Pd(0) (100 mg of resin, containing 22% of Pd). The reaction was kept stirring at 70 °C for 24 h. The reaction was monitored by TLC until complete consumption of the starting product. Once the reaction was finished, the capsule was extracted from the reaction medium and washed (MeOH, water, AcOEt) for reuse. The reaction mixture was diluted with AcOEt (30 mL) and washed with water (10 mL), and the solvent was evaporated in a rotavapor at 50 °C, and the residue was purified on preparative TLC to create compound **32**. 

Compound methyl (E)-3-(1-(4-iodobenzyl)-2,3-dioxoindolin-5-yl)acrylate **32** was obtained as a red solid (60% yield). m.p.= 228–230 °C. ^1^H NMR (CDCl_3_, 400 MHz): 7.79 (d, *J* = 1.7 Hz, 1H), 7.69 (d, J = 8.3 Hz, 1H), 7.69–7.60 (m, 2H), 7.58 (d, *J* = 16.0 Hz, 1H), 7.11 (m, 1H), 7.08 (d, *J* = 8.4 Hz, 1H), 6.78 (d, *J* = 8.2 Hz, 1H), 6.36 (d, *J* = 16.0 Hz, 1H), 4.89 (s, 2H), 3.80 (s, 3H). ^13^C NMR (CDCl_3_, 75.4 MHz MHz): δ 182.3, 166.9, 158.1, 151.7, 142.3, 138.3, 138.0, 133.8, 130.8, 129.3, 124.2, 118.4, 118.1, 111.2, 93.9, 51.9, 43.8. MS (ESI) [M + H]: 448.0. 

In a round bottom flask, 400 mg (1 mmol) of iodoisatin **29** was dissolved in DMF (10 mL). To this solution was added another aqueous solution of sodium acetate (6 mmol in 2 mL of water), methyl acrylate (3 mmol), and Capsule@IRA-120-Pd(0) (100 mg of resin, containing 22% of Pd). The reaction was kept stirring at 90 °C for 12 h. The reaction was monitored by TLC until complete consumption of the starting product. Once the reaction was finished, the capsule was extracted from the reaction medium and washed (MeOH, water, AcOEt) for reuse. The reaction mixture was diluted with AcOEt (30 mL) and washed with water (10 mL), and the solvent was evaporated in a rotavapor at 50 °C and, the residue was purified on preparative TLC to create compound **33**. 

Compound methyl (E)-3-(4-((5-((E)-3-methoxy-3-oxoprop-1-en-1-yl)-2,3-dioxoindolin-1-yl)methyl)phenyl)acrylate **33** was obtained as a red solid (85% yield). m.p. = 188–190 °C. ^1^H NMR (CDCl_3_, 400 MHz): δ 7.80–7.50 (m, 6H), 7.40 (br, 1H), 7.35 (d, *J* = 8.0 Hz, 1H), 6.79 (d, *J* = 8.2 Hz, 1H), 6.44–6.33 (m, 2H), 4.96 (s, 2H), 3.80 (s, 6H). ^13^C NMR (CDCl_3_, 75.4 MHz MHz): δ 182.4, 167.2, 166.9, 158.1, 151.2, 143.7, 142.3, 138.0, 136.2, 134.6, 130.8, 128.8, 127.9, 124.3, 118.6, 118.4, 118.1, 111.2, 51.9, 51.8, 43.9.MS (ESI) [M + H]: 406.0.

### 3.8. Procedure for the Syntheses of Compound ***30***

In a round bottom flask, 400 mg (1 mmol) of 5-iodoisatin **2** was dissolved in DMF (10 mL). To this solution was added another aqueous solution of sodium acetate (3 mmol in 1.5 mL of water), methyl acrylate (3 mmol), and Capsule@IRA-120-Pd(0) (100 mg of resin, containing 22% of Pd). The reaction was kept stirring at 90 °C for 24 h. The reaction was monitored by TLC until complete consumption of the starting product. Once the reaction was finished, the capsule was extracted from the reaction medium and washed (MeOH, water, AcOEt) for reuse. The reaction mixture was diluted with AcOEt (30 mL) and washed with water (10 mL), and the solvent was evaporated in a rotavapor at 50 °C, and the residue was purified on preparative TLC to create compound **30**. 

Compound methyl (E)-3-(1-(3-methoxy-3-oxopropyl)-2,3-dioxoindolin-5-yl)acrylate **31** was obtained as a red solid (85% yield). m.p.= 131–132 °C. ^1^H NMR (CDCl_3_, 400 MHz): δ 7.74–7.71 (m, 2H), 7.58 (d, *J* = 16.0 Hz, 1H), 7.12–7.06 (m, 1H), 6.36 (d, *J* = 16.0 Hz, 1H), 4.03 (t, *J* = 6.8 Hz, 2H), 3.78 (s, 3H), 3.65 (s, 3H), 2.76 (t, *J* = 6.8 Hz, 2H). ^13^C NMR (CDCl_3_, 75.4 MHz MHz): δ 182.4, 171.2, 166.9, 158.2, 151.3, 142.3, 138.0, 130.4, 124.1, 118.1, 117.9, 110.9, 52.1, 51.8, 36.4, 31.9. MS (ESI) [M + H]: 317.29.

### 3.9. Cell Cultures

Cervical HeLa cancer cells and 3T3 mouse fibroblasts from Cell Biolabs (San Diego, CA, USA) were grown in standard culture conditions (5% CO_2_ at 37 °C) in DMEM supplemented with 10% (*v*/*v*) FBS, 2 mM L-glutamine, 1% (*v*/*v*) penicillin/streptomycin, 1 mM sodium pyruvate, and 0.1 mM MEM nonessential amino acids (NEAA).

### 3.10. In Vitro Cell Cytotoxicity

Cytotoxicity of the synthesized isatin-based compounds was tested in vitro by means of the CCK-8 cytotoxicity assay. Cancerous cervical HeLa and 3T3 Balb fibroblasts cells were seeded into 96-well plates (1.0 × 10^4^ cells/well) and grown for 24 h at an optical confluence of 80–90% under standard culture conditions in 100 μL growth medium. Bare cells were used as the negative control. After 24 h of incubation at 37 °C, 100 μL of NPs at 2.5 × 10^10^ NP/mL in the corresponding cell culture medium were injected into the wells and incubated for 24 h and 48 h. After incubation, the culture medium was discarded, cells were washed with 10 mM PBS (pH 7.4) several times, and fresh culture medium (100 μL) containing 10 μL of CCK-8 reagent added to each well. After 2 h, the absorption at 450 nm of cell samples was measured with an UV-vis microplate absorbance reader (Bio-Rad model 689, USA). Cell viability (SR, survival rate) was calculated as follows:$$SR=\frac{Abs_{sample}}{Abs_{blank}}\times 100$$
where *Abs_sample_* is the absorbance at 450 nm for cell samples, and *Abs_blank_* is the absorbance corresponding to the sample controls without the particles.

For the determination of the half-maximal inhibitory concentration (IC_50_), a dose–response curve between the isatin derivatives’ concentrations and percent cell viability was plotted and fitted by means of a nonlinear least-squares fitting method (Microcal Origin 2021) to a four-parameter logistic equation:$$Y=min+\left(\frac{\left(max-min\right)}{1+10^{\left(\left(logIC_{50}-X\right)\times p\right)}}\right)$$
where the original, %control or %survival data are represented by *Y* along their minimal (*min*) and maximal (*max*) values; the isatin derivative concentration is represented by *X*; *IC*_50_ is the concentration at 50% maximal value; and *p* is the slope factor.

## 4. Conclusions

The compartmentalization of solid reagents and catalysts is an interesting strategy that enables not only their easy dispensability and reuse in work-up processes, but also their simultaneous use in chemical reactions, even those in which they are incompatible. Particularly interesting for the pharmaceutical industry is the application of compartmentalized reagents in complex reactions that generate new bioactive heterocyclic structures.

In this work, we have demonstrated for the first time the applicability of the 3D-printing technology in the construction of custom capsule-shaped devices, made of polypropylene, and their integration and application in drug synthesis. These prototypes were tested in SPOS-type reactions as effective systems to compartmentalize supported polymeric reagents, in general, and polymeric catalysts containing metal species, specifically. These devices demonstrated efficacy as containers for the Amberlyst-15 acid reagent as a scavenger of isocyanides in Ugi work-up processes and/or for the IRA-120-Cu(0) and IRA-120-Pd(0) catalysts, in CuAAC- or Heck-type reactions, respectively. These reactions gave rise to the synthesis of different series of diversely substituted isatins, which showed potent activity as antiproliferative agents in HeLa and murine 3T3 Balb fibroblast cell lines. 

## Data Availability

Data is contained within the article and Appendix A.

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
