# Peer review of "3D-Printing of Capsule Devices as Compartmentalization Tools for Supported Reagents in the Search of Antiproliferative Isatins"

_pharmaceuticals, 2023, doi:10.3390/ph16020310_

Round 1

Reviewer 1 Report

It is a timely effort by authors to conduct a study on "3D-printing of capsule devices as compartmentalization tools for supported reagents in the search of antiproliferative isatins". However, there are few concerns which needs to be addressed.

1.  Novelty should be highlighted in a better way.

2. How results and discussion section is coming before meterials and methods, is this not against a norm of general manuscript settings?

3. What were the 3D printing conditions for sample preparations?

4. The glass transition temperature of PP is usually less than zero. But you mentioned it to be higher see line 211 at page 5.

5. Instead we should talk about the melting point of PP! Why did you not perform the DSC for selected PP?

6. How do you define scheme? You may use the word figure!

7. Write those good properties of PP which are more relevant to this current study.

8. Where are the result of SEM and other microscopic examinations? One cannot find them in the manuscript.

Author Response

Moderate English changes were made in the manuscript.

For the rest of questions, please see the attachment

Reviewer 2 Report

Please, see the attached file
